# Current Correlations in a Quantum Dot Ring: A Role of Quantum Interference

**DOI:** 10.3390/e21050527

**Published:** 2019-05-24

**Authors:** Bogdan R. Bułka, Jakub Łuczak

**Affiliations:** Institute of Molecular Physics, Polish Academy of Sciences, ul. M. Smoluchowskiego 17, 60-179 Poznań, Poland

**Keywords:** quantum interference, shot noise, persistent current

## Abstract

We present studies of the electron transport and circular currents induced by the bias voltage and the magnetic flux threading a ring of three quantum dots coupled with two electrodes. Quantum interference of electron waves passing through the states with opposite chirality plays a relevant role in transport, where one can observe Fano resonance with destructive interference. The quantum interference effect is quantitatively described by local bond currents and their correlation functions. Fluctuations of the transport current are characterized by the Lesovik formula for the shot noise, which is a composition of the bond current correlation functions. In the presence of circular currents, the cross-correlation of the bond currents can be very large, but it is negative and compensates for the large positive auto-correlation functions.

## 1. Introduction

In 1985, Webb et al. [1] presented their pioneering experiment, showing Aharonov-Bohm oscillations in a nanoscopic metallic ring and a role of quantum interference (QI) in electron transport. Later, Ji et al. [2] demonstrated the electronic analogue of the optical Mach–Zehnder interferometer (MZI), which was based on closed-geometry transport through single edge states in the quantum Hall regime. Theoretical studies [3,4,5,6,7] predicted coherent transport through single molecules with a ring structure, where, due to their small size, one could show constructive or destructive quantum interference effects at room temperatures. From 2011, these predictions have been experimentally verified, using mechanically controllable break junction (MCBJ) and scanning tunneling microscope break junction (STM-BJ) techniques [8,9] in various molecular systems: Single phenyl, polycyclic aromatic, and conjugated heterocyclic blocks, as well as hydrocarbons (for a recent review on QI in molecular junctions, see [10,11] and the references therein).

Our interest is in the internal local currents and their correlations in a ring geometry to see a role of quantum interference. An interesting aspect is the formation of a quantum vortex flow driven by a net current from the source to the drain electrode, which has been studied in many molecular systems [7,10,12,13,14,15,16,17,18,19,20,21] (see also [22]). It has also been shown that, under some conditions, a circular thermoelectric current can exceed the transport current [23]. In particular, our studies focus on the role of the states with opposite chirality in the ring and on the QI effect and the circular current. Correlations of the electron currents (shot noise) through edge states in the Mach–Zehnder interferometer have been extensively studied by Buttiker et al. [24,25,26,27,28] (see also [29] and the references therein). However, in a metallic (or molecular) ring, the situation is different than in the MZI, as multiple reflections are relevant to the formation of the circular current. Our studies will show that the transition from laminar to vortex flow is manifested in the shot noise of local currents. In particular, it will be seen in a cross-correlation function for the currents in different branches of the ring, which becomes negative and large in the presence of the circular current.

The paper is organized as follows. In the next chapter, Section 2, we will present the model of three quantum dots in a ring geometry, which is the simplest model showing all aspects of QI and current correlations. The model includes a magnetic flux threading the ring, which changes interference conditions as well as inducing a persistent current. The net transport current and the local bond currents, as well as the persistent current (and their conductances), are derived analytically, by means of the non-equilibrium Keldysh Green function technique. It will be shown that the correlation function for the net transport current can be expressed as a composition of the correlation functions for the local currents inside the ring. We will, also, show all shot noise components; in particular, the one for the net transport current (given by Lesovik’s formula [30]). The next chapters, Section 3, Section 4 and Section 5, present the analyses of the results for the case Φ=0 (without the magnetic flux), for the case with the persistent current only (without the source-drain bias *V*), and for the general case (for V≠0, Φ≠0) showing the interplay between the bond currents and the persistent current. Finally, in Section 6, the main results of the paper are summarized.

## 2. Calculations of Currents and Their Correlations in Triangular Quantum Dot System

### 2.1. Model

The considered system of three quantum dots (QDs) in an triangular arrangement is presented in Figure 1. This system is described by the Hamiltonian(1)$$\begin{array}{c} H_{tot}=H_{3QD}+H_{el}+H_{3QD-el}, \end{array}$$which consists of parts corresponding to the electrons in the triangular QD system, in the electrodes, and in the coupling between the sub-systems, respectively. The first part is given by(2)$$\begin{array}{c} H_{3QD}=\sum_{i\in 3QD}\varepsilon_{i}\;c_{i}^{†}c_{i}+\sum_{i,j\in 3QD}\left({\tilde{t}}_{ij}c_{i}^{†}c_{j}+h.c.\right), \end{array}$$where the first term describes the single-level energy, $\varepsilon_{i}$, at the *i*-th QD and the second term corresponds to electron hopping between the QDs. Here, the hopping parameters ${\tilde{t}}_{12}=t_{12}e^{ı\phi /3}={\tilde{t}}_{21}^{*}$, ${\tilde{t}}_{23}=t_{23}e^{ı\phi /3}={\tilde{t}}_{32}^{*}$, and ${\tilde{t}}_{31}=t_{31}e^{ı\phi /3}={\tilde{t}}_{13}^{*}$ include the phase shift ϕ=2πΦ/(hc/e), due to presence of the magnetic flux Φ; where hc/e denotes the one-electron flux quantum. The spin of electrons is irrelevant in our studies and so it is omitted. We consider transport in an open system with the left (L) and right (R) electrodes as reservoirs of electrons, each in thermal equilibrium with a given chemical potential $\mu_{\alpha }$ and temperature $T_{\alpha }$. The corresponding Hamiltonian is(3)$$\begin{array}{cc} H_{el} & =\sum_{k,\alpha \in L,R}\varepsilon_{k,\alpha }\;c_{k\alpha }^{†}c_{k\alpha }, \end{array}$$where $\varepsilon_{k,\alpha }$ denotes an electron spectrum. The coupling between the 3QD system and the electrodes is given by(4)$$\begin{array}{cc} H_{3QD-el} & =\sum_{k}\left(t_{L}\;c_{kL}^{†}c_{1}+t_{R}\;c_{kR}^{†}c_{2}+h.c.\right), \end{array}$$with tunneling from the electrodes given by the hopping parameters $t_{L}$ and $t_{R}$, respectively. The model omits Coulomb interactions and, therefore, one can derive all transport characteristics analytically.

### 2.2. Calculation of Currents

We consider a steady-state current, with the net transport current through the 3QD system, $I^{tr}=I_{12}+I_{13}$, expressed as a sum of the bond currents through the upper and the lower branches(5)$$\begin{array}{c} I_{ij}=\frac{e}{ı\hbar }\left({\tilde{t}}_{ij}\langle c_{i}^{†}c_{j}\rangle -{\tilde{t}}_{ji}\langle c_{j}^{†}c_{i}\rangle \right). \end{array}$$

We use the non-equilibrium Green function technique (NEGF), which is described in many textbooks (e.g., see [31]). To determine the currents, one calculates the lesser Green functions, $G_{ji}^{<}\equiv ı\langle c_{i}^{†}c_{j}\rangle$, by means of the equation of motion method (EOM). The coupling with the electrodes is manifested by the lesser Green functions $g_{\alpha }^{<}=2\pi \left(g_{\alpha }^{r}-g_{\alpha }^{a}\right)f_{\alpha }$, where $g_{\alpha }^{r,a}$ denotes the retarded (r) and advanced (a) Green functions in the α electrode, and $f_{\alpha }=1/\left(\exp\left[\left(E-\mu_{\alpha }\right)/k_{B}T_{\alpha }\right]+1\right)$ is the Fermi distribution function for an electron with energy *E*, with respect to a chemical potential $\mu_{\alpha }$ and at temperature $T_{\alpha }$. For any Green function $G_{ji}^{<}$, we separate contributions from the left and the right electrodes (i.e., we extract the coefficients in front of $g_{L}^{<}$ and $g_{R}^{<}$) and, after some algebra, the bond current can be expressed as(6)$$\begin{array}{cc} I_{ij} & =-\frac{e}{2\pi \hbar }\int_{-\infty }^{\infty }dE\;\left[G_{ij}^{L}\left(E\right)f_{L}-G_{ij}^{R}\left(E\right)f_{R}\right], \end{array}$$where the dimensionless conductances for the upper and the lower branches are(7)$$\begin{array}{cc} G_{12}^{L}= & 2\Gamma_{L}t_{12}ℑ\left[d_{23,31}d_{23,23}^{*}\right]/A, \end{array}$$
(8)$$\begin{array}{cc} G_{12}^{R}= & 2\Gamma_{R}t_{12}ℑ\left[d_{23,31}^{*}d_{31,31}^{*}\right]/A, \end{array}$$
(9)$$\begin{array}{cc} G_{13}^{L}= & 2\Gamma_{L}t_{31}ℑ\left[d_{12,23}d_{23,23}^{*}\right]/A,\;\mathrm{and} \end{array}$$
(10)$$\begin{array}{cc} G_{13}^{R}= & 2\Gamma_{R}t_{31}ℑ\left[d_{12,31}d_{23,31}\right]/A. \end{array}$$Here, we denote the coefficients: $d_{12,23}=e^{-ı\phi }t_{12}t_{23}-t_{31}w_{2}^{r}$, $d_{12,31}=t_{12}t_{31}-e^{-ı\phi }t_{23}w_{1}^{r}$, $d_{23,31}=e^{ı\phi }t_{23}t_{31}-t_{12}w_{3}$, $d_{23,23}=t_{23}^{2}-w_{2}^{r}w_{3}$, $d_{31,31}=t_{31}^{2}-w_{1}^{r}w_{3}$, and the denominator(11)$$\begin{array}{cc} A=|w_{1}^{r}t_{23}^{2}+w_{2}^{r}t_{31}^{2}+w_{3}t_{12}^{2} & -w_{1}^{r}w_{2}^{r}w_{3}-2t_{12}t_{23}t_{31}{\cos\phi |}^{2}, \end{array}$$where $w_{1}^{r,a}=E-\varepsilon_{1}-\gamma_{L}^{r,a}$, $w_{2}^{r,a}=E-\varepsilon_{2}-\gamma_{R}^{r,a}$, $w_{3}=E-\varepsilon_{3}$, $\Gamma_{\alpha }=2ℑ\left[\gamma_{\alpha }^{a}\right]t_{\alpha }^{2}$, and $\gamma_{\alpha }^{r,a}=g_{\alpha }^{r,a}t_{\alpha }^{2}$.

Note that Equation (Equation 6) includes the transport current due to the bias voltage applied to the electrodes, as well as the persistent current induced by the magnetic flux [a term proportional to sinϕ], which can be written as $I_{12}=I_{12}^{tr}-I^{\phi }$ and $I_{13}=I_{13}^{tr}+I^{\phi }$, respectively. These coefficients are coupled with those in (Equation 7)–(Equation 10):(12)$$\begin{array}{c} G_{12}^{L}=G_{12}-G_{\phi }^{L},\;\;G_{12}^{R}=G_{12}+G_{\phi }^{R}, \end{array}$$
(13)$$\begin{array}{c} G_{13}^{L}=G_{13}+G_{\phi }^{L},\;\;G_{13}^{R}=G_{13}-G_{\phi }^{R}. \end{array}$$The first part is(14)$$\begin{array}{cc} I_{ij}^{tr} & =-\frac{e}{2\pi \hbar }\int_{-\infty }^{\infty }dE\left(f_{L}-f_{R}\right)G_{ij}\left(E\right), \end{array}$$where the bond conductances are
(15)$$\begin{array}{cc} G_{12} & =\Gamma_{L}\Gamma_{R}t_{12}w_{3}\left[t_{23}t_{31}\cos\phi -t_{12}w_{3}\right]/A,\;\mathrm{and} \end{array}$$
(16)$$\begin{array}{cc} G_{13} & =\Gamma_{L}\Gamma_{R}t_{23}\left[t_{12}t_{31}w_{3}\cos\phi -t_{23}t_{31}^{2}\right]/A. \end{array}$$The net transport current is $I^{tr}=I_{12}^{tr}+I_{13}^{tr}$ and the transmission is given by(17)$$\begin{array}{cc} T & \equiv G_{12}+G_{13}=\;\Gamma_{L}\Gamma_{R}\left[2t_{12}t_{23}t_{31}w_{3}\cos\phi -t_{12}^{2}w_{3}^{2}-t_{23}^{2}t_{31}^{2}\right]/A. \end{array}$$The persistent current is expressed as(18)$$\begin{array}{c} I^{\phi }\equiv -\frac{e}{\pi \hbar }\;\int_{-\infty }^{\infty }dE\;\left(G_{\phi }^{L}f_{L}+G_{\phi }^{R}f_{R}\right)\;, \end{array}$$where
(19)$$\begin{array}{cc} G_{\phi }^{L}= & \Gamma_{L}t_{12}t_{23}t_{31}\sin\phi \;\left[2t_{23}^{2}-\left(w_{2}^{a}+w_{2}^{r}\right)w_{3}\right]/A,\;\mathrm{and} \end{array}$$
(20)$$\begin{array}{cc} G_{\phi }^{R}= & \Gamma_{R}t_{12}t_{23}t_{31}\sin\phi \;\left[2t_{31}^{2}-\left(w_{1}^{a}+w_{1}^{r}\right)w_{3}\right]/A. \end{array}$$

In the next section, we will show that the voltage bias can induce the circular current, where the bond conductances $G_{ij}$ are larger than unity or negative.

### 2.3. Calculation of Current Correlations

Here, we consider a single-particle interference effect which takes place in a Mach–Zehnder or Michelson interferometer, but not in a Hanbury Brown and Twiss situation with a two-particle interference effect. The current fluctuations are described by the operator $\Delta {\hat{I}}_{ij}\left(t\right)-\langle {\hat{I}}_{ij}\left(t\right)\rangle$, and the current–current correlation function is defined as [32](21)$$\begin{array}{c} S_{ij,nm}\left(t,t^{\prime }\right)\equiv \frac{1}{2}\left[\langle {\hat{I}}_{ij}\left(t\right){\hat{I}}_{nm}\left(t^{\prime }\right)+{\hat{I}}_{nm}\left(t^{\prime }\right){\hat{I}}_{ij}\left(t\right)\rangle -2\langle {\hat{I}}_{ij}(t\rangle \langle {\hat{I}}_{nm}\left(t^{\prime }\right)\rangle \right]. \end{array}$$

We consider the steady currents, for which the correlation functions can be represented, in the frequency domain, by their spectral density(22)$$\begin{array}{c} S_{ij,nm}\left(\omega \right)\equiv 2\int_{-\infty }^{\infty }d\tau e^{ı\omega \tau }S_{ij,nm}\left(\tau \right). \end{array}$$

In this work, we shall restrict ourselves to studying the current correlations at the zero-frequency limit ω=0. As the net transport current is ${\hat{I}}^{tr}={\hat{I}}_{12}+{\hat{I}}_{13}$, its current correlation function can be expressed as a composition of the correlation functions for the bond currents(23)$$\begin{array}{c} S_{tr,tr}=S_{12,12}+S_{13,13}+2S_{12,13}. \end{array}$$The correlation functions $S_{ij,in}$ can be derived by means of Wick’s theorem [31] and are expressed as(24)$$\begin{array}{cc} S_{ij,in}=\frac{e^{2}}{\pi \hbar }\frac{1}{2}\{ & t_{ij}t_{in}\left(\langle c_{i}^{†}c_{j}\rangle \langle c_{i}c_{n}^{†}\rangle +\langle c_{i}^{†}c_{n}\rangle \langle c_{i}c_{j}^{†}\rangle \right))-t_{ij}t_{ni}\left(\langle c_{i}^{†}c_{i}\rangle \langle c_{n}c_{j}^{†}\rangle +\langle c_{n}^{†}c_{j}\rangle \langle c_{i}c_{i}^{†}\rangle \right))\\ + & t_{ji}t_{ni}\left(\langle c_{j}^{†}c_{i}\rangle \langle c_{n}c_{i}^{†}\rangle +\langle c_{n}^{†}c_{i}\rangle \langle c_{j}c_{i}^{†}\rangle \right))-t_{ji}t_{in}\left(\langle c_{i}^{†}c_{i}\rangle \langle c_{j}c_{n}^{†}\rangle +\langle c_{j}^{†}c_{n}\rangle \langle c_{i}c_{i}^{†}\rangle \right))\}. \end{array}$$

Once again, we use the NEGF method. As the lesser Green functions, $G_{ji}^{<}\equiv ı\langle c_{i}^{†}c_{j}\rangle$, and the greater Green functions, $G_{ij}^{>}\equiv -ı\langle c_{i}c_{j}^{†}\rangle$, have the same structure, one should only exchange the Green functions in the electrodes: $g_{\alpha }^{<}=2\pi \left(g_{\alpha }^{r}-g_{\alpha }^{a}\right)f_{\alpha }$↔$g_{\alpha }^{>}=-2\pi \left(g_{\alpha }^{r}-g_{\alpha }^{a}\right)\left(1-f_{\alpha }\right)$. Separating coefficients in front of $f_{L}\left(1-f_{L}\right)$, $f_{R}\left(1-f_{R}\right)$, and $f_{L}\left(1-f_{R}\right)+f_{R}\left(1-f_{L}\right)$, and after some algebra, one can derive a compact formula for any current–current function. The auto-correlation function for the net transport current is given by the well-known Lesovik formula [30,33,34] (see also [32,35] for a multi-terminal and multi-channel case)(25)$$\begin{array}{cc} S_{tr,tr}=\frac{e^{2}}{\pi \hbar } & \int_{-\infty }^{\infty }dE\left\{T^{2}\left[f_{L}\left(1-f_{L}\right)+f_{R}\left(1-f_{R}\right)\right]+T\left(1-T\right)\left[f_{L}\left(1-f_{R}\right)+f_{R}\left(1-f_{L}\right)\right]\right\}, \end{array}$$where T is the transmission through the 3QD system. For a given temperature $T_{L}=T_{R}=T$, one has $f_{L}\left(1-f_{R}\right)+f_{R}\left(1-f_{L}\right)=\coth\left[\left(\mu_{L}-\mu_{R}\right)/2k_{B}T\right]\left(f_{L}-f_{R}\right)$ and, thus,(26)$$\begin{array}{c} S_{tr,tr}=2I^{tr}\coth\left(\frac{eV}{2k_{B}T}\right)-\frac{e^{2}}{\pi \hbar }\int_{-\infty }^{\infty }dE\;T^{2}{\left(f_{L}-f_{R}\right)}^{2}. \end{array}$$When the scale of the energy dependence ΔE of the transmission T is much larger than both the temperature and applied voltage (i.e., $\Delta E\gg eV\gg k_{B}T$), one can obtain the well known explicit relation (see Blanter and Buttiker [32])(27)$$\begin{array}{c} S_{tr,tr}=\frac{e^{2}}{\pi \hbar }\left[2k_{B}T\;T^{2}\left(E_{F}\right)+eV\coth\left(\frac{eV}{2k_{B}T}\right)S_{tr,tr}^{sh}\right]. \end{array}$$

The first term is the Nyquist-Johnson noise at equilibrium and the second term, with $S_{tr,tr}^{sh}=T\left(E_{F}\right)\left(1-T\left(E_{F}\right)\right)$, corresponds to the shot noise [30,32,34].

The correlation functions for the bond currents are calculated from Equation (Equation 24) and are expressed as(28)$$\begin{array}{cc} S_{ij,ik} & =\frac{e^{2}}{\pi \hbar }\int_{-\infty }^{\infty }dE\left\{S_{ij,ik}^{sh}\left[f_{L}\left(1-f_{R}\right)+f_{R}\left(1-f_{L}\right)\right]+\left[G_{ij}^{L}G_{ik}^{L}\;f_{L}\left(1-f_{L}\right)+G_{ij}^{R}G_{ik}^{R}\;f_{R}\left(1-f_{R}\right)\right]\right\}, \end{array}$$where $G_{ij}^{\alpha }$ are given by (Equation 7)–(Equation 10), and the dimensionless spectral functions of the shot noise components are(29)$$\begin{array}{cc} S_{12,12}^{sh} & =\Gamma_{L}\Gamma_{R}t_{12}^{2}{\left|d_{23,31}^{2}-d_{23,23}d_{31,31}^{*}\right|}^{2}/A^{2}, \end{array}$$
(30)$$\begin{array}{cc} S_{13,13}^{sh} & =\Gamma_{L}\Gamma_{R}t_{31}^{2}{\left|d_{12,23}^{*}d_{23,31}^{*}+d_{12,31}d_{23,23}^{*}\right|}^{2}/A^{2}, \end{array}$$
(31)$$\begin{array}{cc} S_{12,13}^{sh} & =\Gamma_{L}\Gamma_{R}t_{12}t_{31}ℜ\left[\left(d_{23,31}^{2}-d_{23,23}d_{31,31}^{*}\right)\left(d_{12,23}^{*}d_{23,31}^{*}+d_{12,31}d_{23,23}^{*}\right)\right]/A^{2},\;\mathrm{and} \end{array}$$
(32)$$\begin{array}{cc} S_{tr,tr}^{sh} & \equiv S_{12,12}^{sh}+S_{13,13}^{sh}+2S_{12,13}^{sh}=\Gamma_{L}\Gamma_{R}{\left|t_{12}\left(d_{23,31}^{2}-d_{23,23}d_{31,31}^{*}\right)+t_{31}\left(d_{12,23}d_{23,31}+d_{12,31}^{*}d_{23,23}\right)\right|}^{2}/A^{2}. \end{array}$$

## 3. Bond Currents and Their Correlations: Driven Circular Current in the Case of Φ=0

Let us analyse the bond currents in detail; first in the absence of the magnetic flux, Φ=0, and for a linear response limit V→0. Using the derivations from the previous section, one can easily calculate the bond conductances and current correlation functions. The results are presented in Figure 2 for an equilateral triangle 3QD system (with all inter-dot hopping parameters $t_{12}=t_{23}=t_{31}=-1$, which is taken as unity in our further calculations) and for various values of the energy level $\varepsilon_{3}$ at the 3rd QD. The central column corresponds to the case $\varepsilon_{1}=\varepsilon_{2}=\varepsilon_{3}=0$, when the eigenenergies are given by $E_{k}=2t\cos k$, for the wave-vector k=0 and the degenerated state for k=±2π/3. It can be seen in the transmission (black curve), which is equal to T=1 at E=−2 and T=0 at E=1, where the Fano resonance takes place, with destructive interference of two electron waves. At low E<0, the incoming wave from the left electrode is split into two branches and the bond conductances are positive, $0\leq G_{12},G_{13}\leq 1$ (see the blue and green curves in the top panel of Figure 2). The cross-correlation function $S_{12,13}^{sh}$ (for the currents in both branches) is positive (see the red curve in the bottom panel in Figure 2). Note that, at the lowest resonant level, all correlation functions $S_{12,12}^{sh}=S_{13,13}^{sh}=S_{12,13}^{sh}=0$, which means that the currents in both branches are uncorrelated.

For E>0, the conductances $G_{12}$ and $G_{13}$ can be negative and exceed unity (with their maximal absolute values inversely proportional to the coupling $\Gamma_{\alpha }$). This manifests a circular current driven by injected electronic waves to the 3QD system, which can not reach the drain electrode; therefore, they are reflected backwards to the other branch of the ring. The circular current can be characterized by the conductance (see also [16])
(33)$$\begin{array}{c} G^{dr}\equiv \left\{\begin{array}{cc} G_{12} & \mathrm{for}\;\;G_{12}<0,\\ -G_{13} & \mathrm{for}\;\;G_{13}<0, \end{array}\right. \end{array}$$where the superscript “*dr*” marks the contribution to the circular current driven by the bias voltage, in order to distinguish it from the persistent current induced by the flux (which will be analysed later). There is some ambiguity in definition of the circular current. Our definition (Equation 33) is similar to the one given by the condition $\mathrm{sign}\left[G_{12}\right]=-\mathrm{sign}\left[G_{13}\right]$ for the vortex flow, used by Jayannavar and Deo [36] and Stefanucci et al. [16] (see [37]–which refers to [7]).

For the considered case in Figure 2b, with $\varepsilon_{3}=0$, the circular current is driven counter-clockwise for 0<E<1 and changes its direction to clockwise at the degeneracy point, E=1 (i.e., when $G_{13}$ becomes negative). All correlation functions are large in the presence of the circular current; their maximum is inversely proportional to $\Gamma_{\alpha }^{2}$. The cross-correlation $S_{12,13}^{sh}$ is large but negative and, therefore, this component reduces the transport shot noise, $S_{tr,tr}^{sh}=S_{12,12}^{sh}+S_{13,13}^{sh}+2S_{12,13}^{sh}$ to the Lesovik formula T(1−T), which reaches zero at the degeneracy point E=1 (see the black curve in Figure 2e). This situation is similar to multi-channel current correlations in transport through a quantum dot connected to magnetic electrodes [38], where cross-correlations for currents of different spins usually reduce the total shot noise to a sub-Poissonian noise with Fano factor F<1 (however, in the presence of Coulomb interactions, the cross-correlations can be positive and lead to a super-Poissonian shot noise with F>1).

The plots on the left and right hand sides of Figure 2 give more insight into the circular current effect. They are calculated for the dot level $\varepsilon_{3}=\mp 2$ shifted by a gate potential, which breaks the symmetry of the system and removes the degeneracy of the states. Three resonant levels can be observed with T=1, where two of them are shifted to the left/right for $\varepsilon_{3}=\mp 2$; however, the state at E=1 is unaffected. There is still mirror symmetry, for which one gets three eigenstates, where two of them are linear compositions of all local states, but the one at E=1 has the eigenvector $1/\sqrt{2}\left(c_{1}^{†}-c_{2}^{†}\right)|0\rangle$, which is separated for the 3rd QD. Therefore, the bond currents are composed of the currents through all three eigenstates, and their contribution depends on *E*. From these plots, one can see that the circular current is driven, for $E>\varepsilon_{3}$, when the cross-correlation $S_{12,13}^{sh}$ becomes negative. The direction of the current depends on the position of the eigenlevels and their current contributions. For $\varepsilon_{3}=-2$, the current circulates clockwise, whereas its direction is counter-clockwise for $\varepsilon_{3}=2$.

Here, we assumed a flat band approximation (FBA) for the electronic structure in the electrodes (i.e., the Green functions $g_{\alpha }^{r,a}=\mp ı\pi \rho$, where ρ denotes the density of states). Appendix A presents analytical results for the currents and shot noise in the fully-symmetric 3QD system coupled to a semi-infinite chain of atoms. The results are qualitatively similar. However, the FBA is more convenient for the analysis than the system coupled to atomic chains; in particular, for the cases with $\varepsilon_{3}=\mp 2$, when localized states appear at −2.99 and 2.56 (i.e., below/above the energy band of the atomic chain).

The above analysis was performed under the assumption of a smooth energy dependence of the conductance in the small voltage limit V→0 and at T=0. However, the conductances exhibit sharp resonant characteristics in the energy scale $\Delta E\propto \Gamma_{\alpha }$ and, therefore, one can expect that these features will be smoothed out with an increase of voltage bias and temperature. Figure 3 presents the Fano factor $F=S_{tr,tr}/2eI^{tr}$, which is the ratio of the current correlation function to the net transport current, which was calculated numerically from Equations (Equation 17) and (Equation 26). At E=−2, one can observe the evolution from the coherent regime, from F=0 to F=1/2 in the sequential regime, for $eV\gg \Gamma_{\alpha }$ or $k_{B}T\gg \Gamma_{\alpha }$. Quantum interference plays a crucial role at E=1, leading to the Fano resonance for which the transmission T=0 and F=1 in the low voltage/temperature regime. An increase of the voltage/temperature results only in a small reduction of the Fano factor.

## 4. Persistent Current and Its Noise: The Case V=0

The persistent current and its noise has been studied in many papers (e.g., by Büttiker et al. [39,40,41,42], Semenov and Zaikin [43,44,45,46], Moskalates [47], and, more recently, by Komnik and Langhanke [48]) using full counting statistics (FCS), as well as in 1D Hubbard rings by exact diagonalization by Saha and Maiti [49] (see, also, the book by Imry [50]).

Here, we briefly present the results for the persistent current and shot noise in the triangle of 3QDs. Notice that, in the considered case, the phase coherence length of electrons is assumed to be larger than the ring circumference, $L_{\phi }\gg L$ [51]. The circular current is given by Equation (Equation 18), which shows that all electrons, up to the chemical potential in the electrodes, are driven by the magnetic flux Φ. Figure 4 exhibits the plots of $I^{\phi }$, derived from Equation (Equation 18), for different couplings with the electrodes. In the weak coupling limit, where Γ→0 and the perfect ring is embedded in the reservoir, the persistent current can be simply expressed as
(34)$$\begin{array}{c} I^{\phi }=e\sum_{k}v_{k}f_{k}=-\frac{e}{\hbar }\sum_{k}2t\sin\left(k+\phi /3\right)f_{k}\;, \end{array}$$where $f_{k}=1/\left(\exp\left[\left(E_{F}-E_{k}\right)/k_{B}T\right]+1\right)$ is the Fermi distribution for the electron with wave-vector *k*, energy $E_{k}=2t\cos\left(k+\phi /3\right)$, and velocity $v_{k}=\left(1/\hbar \right)\partial E_{k}/\partial k=\left(-2t/\hbar \right)\sin\left(k+\phi /3\right)$, and where ϕ=2πΦ/(hc/e) is the phase shift due to the magnetic flux Φ. The sum runs over k=2πn/(Na) for n=0,±1, where N=3 and a=1 is the distance between the sites in the triangle. The current correlator is derived from Equation (Equation 24)
(35)$$\begin{array}{c} S_{\phi ,\phi }=\frac{e^{2}}{\hbar }\sum_{k}4t^{2}\sin^{2}\left(k+\phi /3\right)f_{k}\left(1-f_{k}\right)\;. \end{array}$$

This result says that fluctuations of the persistent current could occur when the number of electrons in the ring fluctuates (i.e., an electron state moves through the Fermi level and $I^{\phi }$ jumps). We show, below, that the coupling with the electrodes (as a dissipative environment) results in current fluctuations [40,41], as well.

At the limit, V→0, the integrand function of the noise $S_{ij,in}$, Equation (Equation 28), is proportional to f(E)(1−f(E)), which becomes the Dirac delta for T→0 and, therefore, one can analyze the spectral function $S_{\phi ,\phi }=S_{12,12}+S_{13,13}-2S_{12,13}$, where the components are $S_{ij,in}=S_{ij,in}^{sh}+G_{ij}^{L}G_{in}^{L}+G_{ij}^{R}G_{in}^{R}$ (see Equations (Equation 7)–(Equation 10) and (Equation 29)–(Equation 31)). Figure 5 presents the correlation function $S_{\phi ,\phi }$ and its various components for the Fermi energy $E_{F}=-1.5$ and the strong coupling $\Gamma_{L}=\Gamma_{R}=1$ when fluctuations are large. Notice that the fluctuations of the bond currents $S_{12,12}$ and $S_{13,13}$ (the blue and green curves, respectively) are different, although the average currents are equal. The cross-correlation function $S_{12,13}$ is positive at ϕ=0, but it becomes negative for larger ϕ, due to the quantum interference between electron waves passing through different states (as described in the previous section).

Figure 5 also shows ${\left(G_{12}^{L}\right)}^{2}$ (blue-dashed curve) and ${\left(G_{12}^{L}\right)}^{2}$ (blue-dotted curve), which correspond to the local fluctuations of the injected/ejected currents to/from the upper branch on the left and right junctions, respectively (see Equation (Equation 28)). The magnetic flux breaks the symmetry, inducing the persistent current and, therefore, the local conductances $G_{12}^{L}$ and $G_{12}^{R}$ are asymmetric.

## 5. Correlation of Persistent and Transport Currents, Φ≠0 and V≠0

In this section, we analyze the currents and their correlations in the general case, derived from Equations (Equation 6), (Equation 14), (Equation 18), and (Equation 28), in the presence of voltage bias and magnetic flux. The results for the conductances and the spectral functions of the shot noise are presented in Figure 6. The magnetic flux splits the degenerated levels at E=1 and destroys the Fano resonance. Figure 6a shows that there is no destructive interference for a small flux ϕ=2π/16, and the transmission is T=1 for all resonances. One can observe the driven circular current for E>0, with negative $G_{12}$ and $G_{13}$, but their amplitudes are much lower than in the absence of the flux (compare with Figure 2b for ϕ=0). For a larger flux, ϕ=2π/4, there is no driven component of the circular current (see Figure 6b, where $G_{12},\;G_{13}\geq 0$). It can also be seen that, for the state at E=0, the electronic waves pass only through the lower branch of the ring, and the upper branch is blocked (with $G_{13}=1$ and $G_{12}=0$, respectively).

The lower panel of Figure 6 presents the spectral functions of the shot noise. According the Lesovik formula, $S_{tr,tr}^{sh}=0$ at the resonant states (as T=1). This seems to be similar to the case ϕ=0 presented in the lower panel in Figure 2. However, there is a great difference in the components of the shot noise $S_{ij,in}^{sh}$, indicating the different nature of transport through these states and the role of quantum interference. Let us focus on the lowest resonant state, at E=−2, in Figure 6c, and compare with that in Figure 2e, in the absence of the flux. In the former case, the currents in both branches were uncorrelated, and $S_{12,12}^{sh}=S_{13,13}^{sh}=S_{12,13}^{sh}=0$. In the presence of the flux, quantum interference becomes relevant, which is seen in the shot noise (Figure 6c). Now, the currents in both branches are correlated; $S_{12,13}^{sh}$ is negative close to resonance and fully compensates for the positive contributions $S_{12,12}^{sh}$ and $S_{13,13}^{sh}$ at resonance. For ϕ=2π/4 (see Figure 6d), all shot noise components are large, which indicates a strong quantum interference effect.

Figure 7 shows the Fano factor in the presence of the flux ϕ=2π/16 and for various bias voltages. Compared with the results in Figure 3 for ϕ=0, one can see how a small flux can destroy quantum interference and change electron transport. It is particularly seen close to E=−1, where the states with opposite chirality are located. In the case ϕ=0, one can observe the Fano resonance with a perfect destructive interference, T=0 and F=1. With an increase of the flux ϕ, the Fano dip disappears, the two states are split, and transmission reaches its maximum value T=1; the Fano factor F=0 when the splitting $\Delta E>\Gamma_{\alpha }$. A similar effect was seen in the case of Figure 2, where a change of the position of the local level $\varepsilon_{3}$ removed the state degeneracy and destroyed the Fano resonance.

For the strong coupling $\Gamma_{L}=\Gamma_{R}=1$, the intensity of the transport current is comparable to the persistent current and, therefore, one can expect a significant driven circular current. Figure 8 presents the flux dependence of the total circular current $I^{c}$ and its driven component $I^{dr}$, as well as the transport current $I^{tr}$, for various voltages. For the considered case $E_{F}=0.9$, the driven current circulates counter-clockwise and deforms the flux dependence of the circular currents, which become asymmetric.

## 6. Summary

We considered the influence of quantum interference on electron transport and current correlations in a ring of three quantum dots threaded by a magnetic flux. We assumed non-interacting electrons and calculated the bond conductances, the local currents, and the current correlation functions—in particular, the shot noise—by means of the non-equilibrium Keldysh Green function technique, taking into account multiple reflections of the electron wave inside the ring. As we considered elastic scatterings, for which Kirchhoff’s current law is fulfilled, the transmission $T=G_{12}+G_{13}$ is a sum of the local bond conductances and the shot noise for the transport current is a composition of the local current correlation functions, $S_{tr,tr}^{sh}=S_{12,12}^{sh}+S_{13,13}^{sh}+2S_{12,13}^{sh}=T\left(1-T\right)$, which gives the Lesovik formula.

In the system, having triangular symmetry, the eigenstates $E_{0}=-2$ and $E_{\pm }=1$ (with the wavevector k=0 and k=±2π/3) play a different role in the transport, which is seen in the bond conductances and the shot noise components. An electron wave injected with energy close to $E_{0}$ is perfectly split into both branches of the ring and the current cross-correlation function $S_{12,13}^{sh}$ is positive. At the resonance $E_{0}$, the transmission T=1 and all correlation functions $S_{12,12}^{sh}=S_{13,13}^{sh}=S_{12,13}^{sh}=0$, which means that the bond currents are uncorrelated. The magnetic flux changes quantum interference conditions and correlates the bond currents; the cross-correlation $S_{12,13}^{sh}$ becomes negative at the resonance and fully compensates the positive auto-correlation components $S_{12,12}^{sh}$ and $S_{13,13}^{sh}$ (with $S_{tr,tr}^{sh}=0$).

Quantum interference plays a crucial role in transport through the degenerate states at $E_{\pm }=1$, where one can observe Fano resonance with destructive interference. In this region, the circular current $I^{dr}$ can be driven by the bias voltage. The bond conductances have an opposite sign, their maximal value is inversely proportional to the coupling, $\Gamma_{\alpha }$, with the electrodes, and they can be larger than unity. The direction of $I^{dr}$ depends on the bias voltage and the position of the Fermi energy $E_{F}$, with respect to the degenerate state $E_{\pm }$. The auto-correlation functions $S_{12,12}^{sh}$, $S_{13,13}^{sh}$ are large (inversely proportional to $\Gamma_{\alpha }^{2}$) close to the resonance. The cross-correlator $S_{12,13}^{sh}$ is negative in the presence of the driven circular current. Our calculations show that a small magnetic flux, ϕ=2π/16, can destroy the Fano resonance, and two resonance peaks (with T=1) appear. The driven component, $I^{dr}$, is reduced with an increase of ϕ, and it disappears at ϕ=2π/4. However, quantum interference still plays a role; the bond currents are strongly correlated (with large $S_{12,12}^{sh}$ and $S_{13,13}^{sh}$ and negative $S_{12,13}^{sh}$). For a large coupling, the driven part $I^{dr}$ can be large and can profoundly modify the total circular current $I^{c}=I^{dr}+I^{\phi }$.

We also performed calculations of the bond currents and their correlations for rings with a various number of sites; in particular, for the benzene ring in para-, metha-, and ortho-connection with the electrodes. The results are qualitatively similar to those presented above for the 3QD ring: Quantum interference of the travelling waves with the eigenstates of opposite chirality leads to the driven circular currents, accompanied by large current fluctuations with a negative cross-correlation component. To observe this effect, the two conducting branches should be asymmetric; in particular, in the benzene ring, the driven circular current appears for the metha- and ortho-connections, but is absent in the para-connection, where both conducting branches are equivalent (see also [7]).

An open problem is including interactions between electrons into the calculations of the coherent transport and shot noise. Coulomb interactions can be taken into account in the sequential regime [52], or by using the real-time diagrammatic technique [53,54,55]; however, in practice, one includes only first- and second-order diagrams with respect to the tunnel coupling and the role of QI is diminished. In principle, one can treat QI on an equal footing with electron interactions in the framework of quantum field theory [56], as was done for the Anderson single impurity model, by means of full counting statistics (FCS), where the average current and all its moments were calculated [57]. However, this is a formidable task, even for the simple 3QD model.

## Figures and Tables

**Figure 1 entropy-21-00527-f001:**
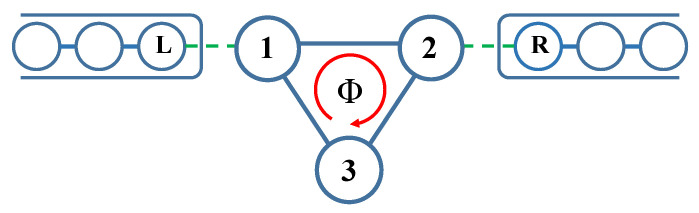
Model of the triangular system of three quantum dots (3QDs) threaded by the magnetic flux Φ and attached to the left (L) and the right (R) electrodes.

**Figure 2 entropy-21-00527-f002:**
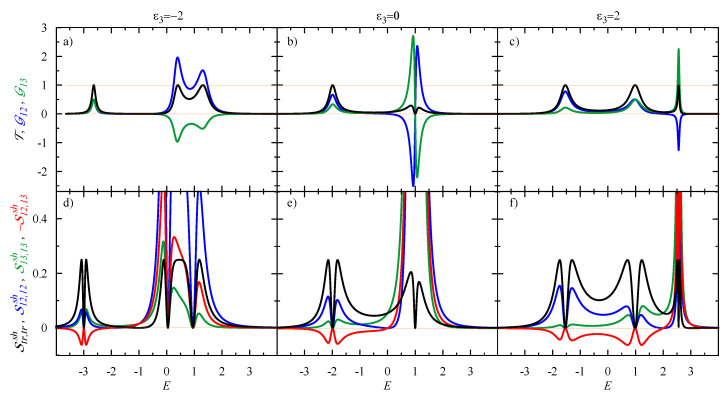
(Top) Transmission and dimensionless bond conductances: T—black, $G_{12}$—blue, and $G_{13}$—green. (Bottom) Dimensionless spectral function of the shot noise: $S_{tr,tr}^{sh}$—black, $S_{12,12}^{sh}$—blue, $S_{13,13}^{sh}$—green, and $-S_{12,13}^{sh}$—red; calculated as a function of the electron energy *E* for the equilateral triangle system of 3QDs (with the inter-dot hopping $t_{12}=t_{23}=t_{31}=-1$, which is taken as unity in this paper) in the linear response limit V→0. The dot levels are $\varepsilon_{1}=\varepsilon_{2}=0$ and $\varepsilon_{3}=-2$, 0, 2, for left, center, and right columns, respectively. The coupling with the electrodes is taken to be $\Gamma_{L}=\Gamma_{R}=0.25$. Note that the cross-correlation function $S_{12,13}^{sh}$ (red) is plotted negatively to show the zero crossing more clearly.

**Figure 3 entropy-21-00527-f003:**
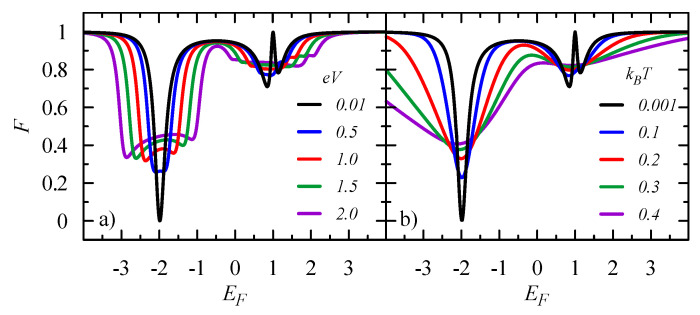
Fano factor as a function of the Fermi energy $E_{F}$ for the equilateral triangle 3QDs system ($t_{12}=t_{23}=t_{31}=-1$ and $\varepsilon_{1}=\varepsilon_{2}=\varepsilon_{3}=0$) (**a**) for various bias voltages eV=0.01, 0.5, 1.0, 1.5, and 2.0, at T=0; and (**b**) for various temperatures $k_{B}T=0.001$, 0.1, 0.2, 0.3, and 0.4, for V→0. The coupling to the electrodes is taken as $\Gamma_{L}=\Gamma_{R}=0.25$, and the chemical potentials in the electrodes are $\mu_{L}=E_{F}-eV/2$ and $\mu_{R}=E_{F}+eV/2$.

**Figure 4 entropy-21-00527-f004:**
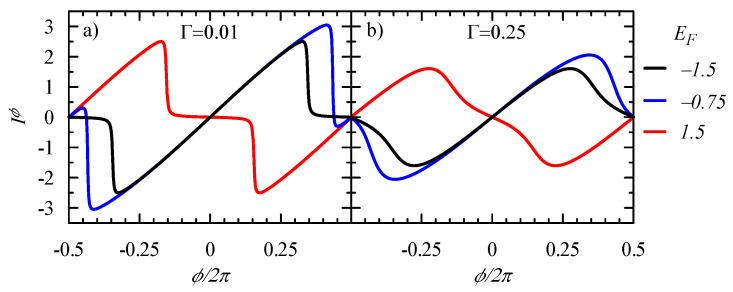
Persistent current $I^{\phi }$ versus the flux ϕ threading the equilateral triangle system of 3QDs ($t_{12}=t_{23}=t_{31}=-1$ and $\varepsilon_{1}=\varepsilon_{2}=\varepsilon_{3}=0$). The coupling is taken as $\Gamma_{L}=\Gamma_{R}=\Gamma =0.01$ and 0.25; the Fermi energies are $E_{F}$= −1.5 (**black**), −0.75 (**blue**), 1.5 (**red**); and T=0.

**Figure 5 entropy-21-00527-f005:**
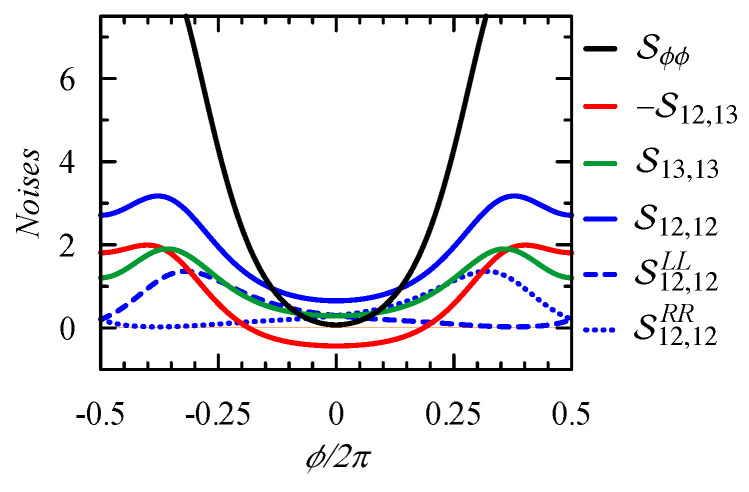
Flux dependence of spectral function of the persistent current correlator $S_{\phi ,\phi }$ (**black**) and its components: $S_{12,12}$ (**blue**), $S_{13,13}$ (**green**), -$S_{12,13}$ (**red**), and $S_{12,12}^{LL}={\left(G_{12}^{L}\right)}^{2}$ (**blue-dashed**), $S_{12,12}^{RR}={\left(G_{12}^{R}\right)}^{2}$, (**blue-dotted**), respectively. We assume strong coupling: $\Gamma_{L}=\Gamma_{R}=1.0$, $E_{F}=-1.5$, and T=0.

**Figure 6 entropy-21-00527-f006:**
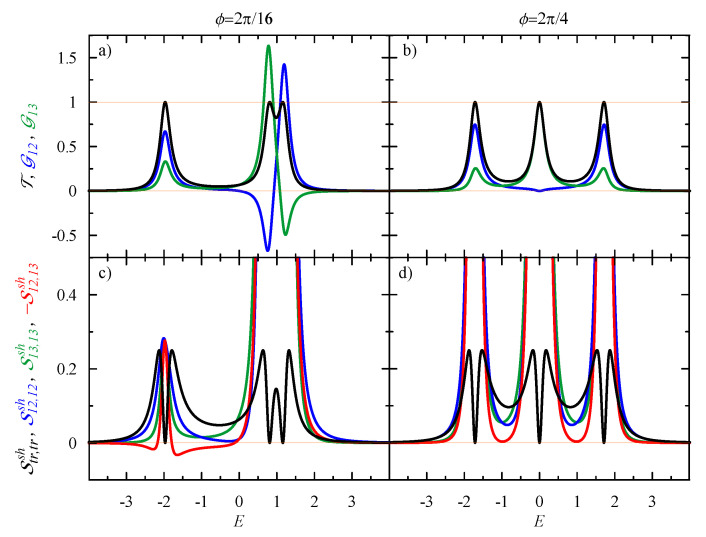
(**Top**) Energy dependence of driven conductance $G_{12}$ (blue), $G_{13}$ (**green**) and transmission T (**black**). (**Bottom**) Shot noise $S_{tr,tr}^{sh}$ (**black**) with the components: $S_{12,12}^{sh}$ (**blue**), $S_{13,13}^{sh}$ (**green**), and $-S_{12,13}^{sh}$ (**red**) for the considered triangular 3QD system threaded by the flux ϕ=2π/16 (**left**) or ϕ=2π/4 (**right**); the coupling is $\Gamma_{L}=\Gamma_{R}=0.25$, and T=0. Note that we plot $-S_{12,13}^{sh}$.

**Figure 7 entropy-21-00527-f007:**
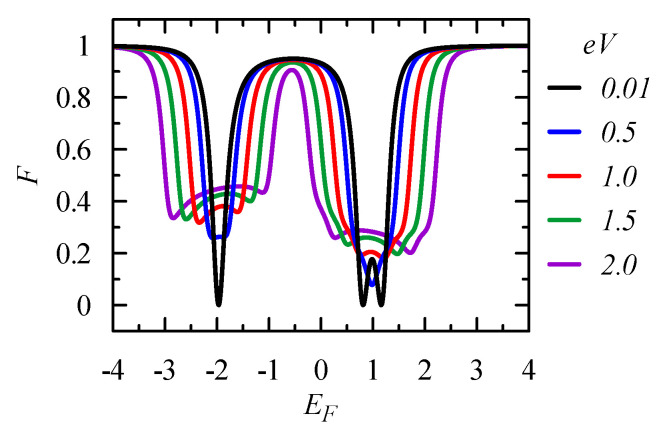
Fano factor as a function of $E_{F}$ for the considered 3QD system threaded by the flux ϕ=2π/16 and for various bias voltages eV=0.01, 0.5, 1.0, 1.5, and 2.0. The coupling is $\Gamma_{L}=\Gamma_{R}=0.25$, the chemical potentials are $\mu_{L}=E_{F}-eV/2$ and $\mu_{R}=E_{F}+eV/2$, and T=0.

**Figure 8 entropy-21-00527-f008:**
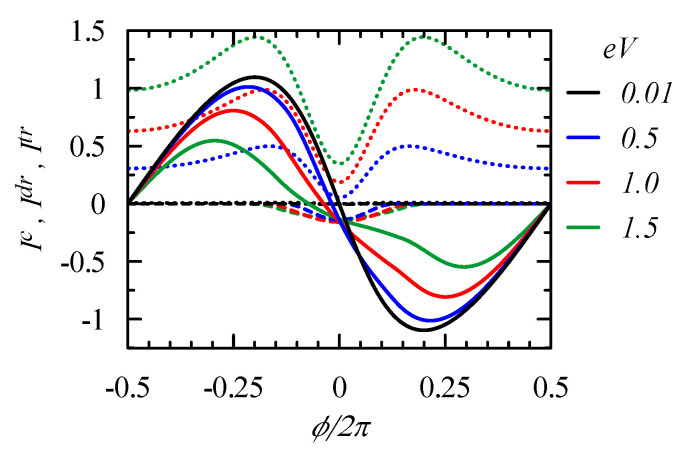
Circular current $I^{c}=I^{dr}+I^{\phi }$ (solid curves), its driven component $I^{dr}$ (dashed curves), and the net transport current $I^{tr}$ (dotted curves) versus ϕ for various bias voltages: eV=0.01, 0.5, 1.0, and 1.5. We assume a strong coupling $\Gamma_{L}=\Gamma_{R}=1$, the chemical potentials are $\mu_{L}=E_{F}-eV/2$, $\mu_{R}=E_{F}+eV/2$, $E_{F}=0.9$, and T=0.

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
