# Peer review of "Current Correlations in a Quantum Dot Ring: A Role of Quantum Interference"

_entropy, 2019, doi:10.3390/e21050527_

Round 1

Reviewer 1 Report

The paper concerns the transport properties, in particular the current correlations, of a system consisting of a ring (composed of three quantum dots) attached to two terminals and in the presence of a magnetic flux threading the ring. Aiming at investigating the effect of interference, the authors use a fully coherent (non-interacting) theoretical approach to derive expressions for the current and noise in the various branches of the system in the presence of a bias voltage and/or a magnetic flux. The authors study the energy-dependence of the quantities that compose current and noise, and finally investigate the latter two as functions of the flux in order to single out the effect of interference, such as the occurrence of Fano resonances.

The paper is sound, clearly written and the results are interesting to the field of quantum transport in mesoscopic systems. In conclusion, the paper deserves publication in Entropy.

Minor comments:

- Fig. 2,3, 5, 6, 7: in which units the shot noise and Fano factors are plotted?

- In the text which describes Fig. 5, the author states that the noise S_{phi,phi} is maximal at phi=+-0.36. According to the plot, however, such noise seems to be always increasing. Can the author clarify this point?

Author Response

The reviewer comment:

- Fig. 2,3, 5, 6, 7: in which units the shot noise and Fano factors are plotted?

Our response:

All plots are made for the dimensionless conductances and the dimensionless spectral functions of the shot noise. In front of Eqs.(29)-(32) which define the components of the shot noise, we added the text (line 112): “the dimensionless spectral function”. More, we added the text (line 118):  “all inter-dot hopping parameters $t_{12}=t_{23}=t_{31}=-1$, which is taken as the unity in our further calculations”. Of course the Fano factor is dimensionless by its definition.

The reviewer comment:

- In the text which describes Fig. 5, the author states that the noise S_{phi,phi} is maximal at phi=+-0.36. According to the plot, however, such noise seems to be always increasing. Can the author clarify this point?

Our response:

In Fig.5 the maximum of S_{phi,phi} is beyond the plot, because we wanted to focus on the other shot noise components with much lower values. To make the text clear we removed this sentence, which is not so relevant.

Reviewer 2 Report

Authors present a thorough analysis of the coherence effects in the electron transport trough single ring-shaped molecules. Spin and Coulomb interactions are not included in the model, which, however allows  to derive analytically the relevant transport quantities in terms of the Green's functions formalism.  The formalism can basically treat stationary transport and thus effects in steady state currents. However this study is important as it provides the base for a future study of many body and transient effects. The paper is not self-contained, but written with experience, the list of references is exhaustive.

Author Response

We appreciate the suggestion of the referee to study of many body and transient effects in the future.

English was polished.